# Growth and Microstructural Features in Otoliths of Larval and Juvenile *Sinogastromyzon wui* (F. Balitoridae, River Loaches) of the Upper Pearl River, China

Minghui Gao [1], Zhiqiang Wu [2,*], Liangliang Huang [2,3,*], Xichang Tan [4], Mingsi Li [2] and Haibo Huang [2]

1   College of Life Science and Technology, Guangxi University, Nanning 350000, China; gmhxxzj@126.com
2   College of Environmental Science and Engineering, Guilin University of Technology, Guilin 541000, China; lms980330@163.com (M.L.); eleveshhb@126.com (H.H.)
3   Coordinated Innovation Center of Water Pollution Control and Water Security in Karst Area, Guilin University of Technology, Guilin 541004, China
4   Bureau of Hydrology and Water Resources, Pearl River Conservancy Commission of Ministry of Water Resources, Guangzhou 510000, China; rjimtxc@hotmail.com
*   Correspondence: wuzhiqiang@glut.edu.cn (Z.W.); llhuang@glut.edu.cn (L.H.)

**Abstract:** Otolith growth and microstructural features of fish are essential to the understanding of the early fish lifecycle. This paper assesses the features of otoliths from laboratory-reared larval and juvenile *Sinogastromyzon wui* (*S. wui*, 0 to 25 days post-hatching) that were obtained as eggs from the Shilong Reach of Xijiang River between April and August 2021. We observed the development of the three pairs of otoliths (lapilli, sagittae, and asterisci) and compared the shape changes and growth of the lapilli and sagittae, as well as the timing and deposition rate of increments of the lapilli. The lapilli and the sagittae were visible on hatching, whereas the asterisci were present at four days post-hatching (dph). The shape of the sagitta changed more obviously than that of the lapillus, and a strong correlation was observed between sagitta shape changes and fish ontogenesis. The otolith shape greatly modulated during the post-flexion larval stage (Post-FLS), it corresponded with the formation period of individual fins. Analysis of the microstructural features indicated that lapilli were the optimal otolith for age determination and increment deposition rate confirmation. Using regression analysis of the known age and the number of lapillus daily increments, we demonstrated that the lapillus developmental increments were deposited daily, and the first increment formed at two days post-hatching. Our conclusions support employing the lapillus increment deposition rate and the time of the first daily increments in the determination of the age of wild larval and juvenile *S. wui*.

**Keywords:** ontogeny; teleost anatomy; development; fish morphology; Fourier

## 1. Introduction

Otoliths are acellular biomineralized concretions of calcium carbonate and other minor elements (Na, Sr, K, S, N, Cl, and P), generated on a protein matrix in vertebrates' inner ears [1]. These structures are mainly used for sound reception and balance orientation [2,3]. The evidence of daily otolith growth was discovered and described by Panella in the 1970s [4]. In subsequent studies, the otolith daily increment deposition was shown to commonly occur in the early life stages of Osteichthyes [5,6].

The fish developmental rule involves resource dynamic assessment and fishery management [7,8]. During the early fish lifecycle, otoliths develop with the growing fish. The otolith growth model is based on the otolith radius (OR) and the total length (TL), which are used to infer the developmental rate of the larval and juvenile fish [9]. The daily increment width is influenced by water temperature and prey and can be used to study the spawning populations in varying seasons [10,11]. In addition, the otolith shape alters as it grows. Therefore, otolith shape analysis is often used to distinguish fish species or populations [12,13].

Otoliths are one of the most studied elements of the teleost fish anatomy, because they represent a permanent record of life history [14]. Otolith development is modulated by both endogenous (ontogeny, physiology, and feeding habits) and exogenous elements (water depth, temperature, salinity, and substrate), and the otolith microstructure possesses a certain degree of species-specificity [15,16]. Hence, the otolith microstructure is often used to examine early life events such as hatching, first-feeding, and habitat alteration [17,18]. In addition, the central nuclear characteristics can be used to distinguish between varying fish populations [19], while the width of the otolith increments can be used to analyze the growth stage of fish in early life history [20].

*Sinogastromyzon wui* comes from the Balitoridae family and Homalopterinae subfamily. It is among the most prevalent species in the upper reaches of the Pearl River [21]. *S. wui* spawns drifting eggs, which develop in drifting water and it is a member of Ostariophysi, whose inner ear (and sagittae, in particular) is highly modified during their ontogenesis. In recent years, with the dams' construction, the river's hydrology has altered significantly. The reproduction and population recruitment of *S. wui* are key indices for the evaluation of fisheries' resources. However, there are limited studies on the early growth of *S. wui*. Herein, we explored the morphological alterations in otoliths during ontogenesis and examined a potential association between age and shape formation. Furthermore, we confirmed the deposition time of the first increment and identified the deposition frequency. Our conclusions will provide important benefits to fishery resource protection and assessment.

## 2. Materials and Methods

### 2.1. Fish Rearing and Sampling

Our fish egg samples were collected from the Shilong Reach of the Xijiang River (109°31′30″ E, 23°52′21″ N) between April and August 2021. The collection was performed daily during the spawning season using Jiang nets (total length 5 m; rectangular iron opening/mouth 1.0 m × 1.5 m and a mesh net size of 0.5 mm attached to a 0.8 m × 0.4 m × 0.4 m filter collection bucket), which were placed vertically below the surface of the water and against the current. Each sampling period lasted 1 h (6–7 a.m.).

After collection, the eggs were sorted in the laboratory. *S. wui* eggs were readily recognized by their non-viscous pale yellow texture, and they were of a radius between 0.4 and 0.5 cm. Each sample collection rendered 50–80 *S. wui* eggs, which were then maintained in 1.5 L Zug bottles. Once the eggs hatched, the larvae were transferred to 30 L water pools. The water in the Zug bottles and pools was river water from the sampling site, and an air distributor was added to provide sufficient oxygen. The water temperature was maintained between 19 and 28 °C. The photoperiod was equal to the natural length of the day (23°52′ N). Following 2 dph, the larvae were fed with cooked chicken egg yolks three times a day. After half an hour, any unconsumed yolks were suctioned out of tanks, with the help of a water cleaner (WEIPRO, TC3500, Guangzhou, China), to keep the water clean.

### 2.2. Otolith Preparation

While rearing, all samples were separated into four ontogenetic stages: the preflexion larval stage (Pre-FLS), the flexion larval stage (FLS), the post-flexion larval stage (Post-FLS), and the juvenile stage (JS) [22]. Otolith samples were collected daily during Pre-FLS and FLS; then, that otolith samples were selected once every 2 days during Post-FLS and JS. In total, 320 individuals ranging from 5.66 to 13.93 mm were sampled (10 in every age group used for shape research and 10 for validating increment deposition frequency).

Before dissection, the larvae and juveniles were anesthetized with lidocaine hydrochloride (30 mg/L), and the TL was measured with vernier calipers. All of the otoliths (lapilli, sagittae, and asterisci) from the experimental fish were selected using a microscope (Motic, SMZ-168). The otoliths were first cleaned with absolute ethyl alcohol and, subsequently, encapsulated on a glass slide with colorless enamel resin. Finally, the otoliths were photographed under a microscope (Olympus, BX53) with a camera (MicroPublisher 5.0 Real-Time Viewing).

### 2.3. Assessment of Otolith Development

Given that asterisci were not present upon hatching, this study only examined lapilli and sagittae growth. Otoliths are generally divided into four growth areas; namely, anterior, posterior, dorsal, and ventral (Figure 1) [23]. The growth rate of each area was expressed by the otolith radius of that area. Moreover, the otolith radius was measured using a computer image analysis system (computer imaging identification system for the analysis of otolith microstructure of fish) [24].

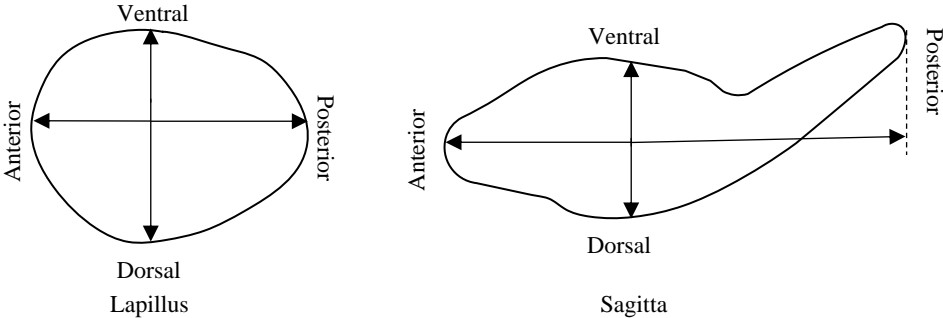

**Figure 1.** The measured radius of each growth area of lapillus and sagitta.

To better elucidate otolith shape alterations with age, parameters closely linked to 140 groups of lapilli and sagittae (once every 2 days during the larval and juvenile stages) were obtained using elliptic Fourier analysis [25,26]. The shapes of lapilli and sagittae were described as a two-dimensional projection carrying elements termed harmonics. Next, each otolith was represented by four elliptic Fourier coefficients, which mathematically normalized its shape, without any alterations to the otolith size, position, or rotation. Following this, the Fourier power spectrum was calculated for each otolith, which determined the appropriate quantity of harmonics for each otolith using a 99.99% accurate contour definition [27]. Our results suggested 20 harmonics, with each harmonic carrying 4 coefficients, namely, a, b, c, and d (80 Fourier coefficients in total). However, since the initial three elliptic Fourier coefficients of the first harmonics were adjusted to constant values (a = 1, b = 0, and c = 0), they were disregarded, giving us a final count of 77 Fourier coefficients. Standardization and coefficient calculations were carried out in SHAPE 1.3 software [28].

### 2.4. Daily Increment Verification and Spacing Assessment

We excluded sagittae from the daily increment assessment due to its highly fragile nature. Instead, we employed 25 dph left lapilli, each of which was encapsulated on a glass slide with colorless enamel resin and buffed using 3000-grit sandpapers to expose the central nucleus. Images were captured with a digital camera, as described before. Lapillus counting was conducted twice by three separate individuals. Error rates were calculated for each counter. Readings were only included in the analysis if the error rate remained <5% [26].

### 2.5. Statistical Analysis

To determine whether the otolith shape differed between ontogenetic stages, a canonical discriminant analysis (CDA) on the principal component scores in the scores file (*.pcs) output by PrinComp was performed in SPSS 2020. The Wilks' Lambda of each discriminant function was used to assess the CDA performance, ranging from 0 (total discrimination) to 1 (no discrimination).

The linear relationship between the known age and daily increment quantity was used to determine the deposition frequency. A *t*-test was used to test the difference between the slope of the linear function and 1. The increment frequency was verified when the t between the slope and 1 was insignificant. The association between the otolith radius (OR) and age (days) was fitted to the following logistic growth model: $OR_t = (OR_\infty \times OR_0)/((OR_\infty - OR_0)\, e^{-kt} + OR_0)$, gompertz growth model: $OR_\infty \times (OR_\infty/OR_0)\hat{\ }e^{-kt}$, or linear growth model: $OR_t = kt + OR_0$.

In both cases, t represented age in days, $OR_t$ represented the OR stage *t*, *k* represented the instantaneous growth rate, and $OR_0$ represented the initial *OR*. The parameters were derived using Origin 2018 (OriginLab, Northampton, MA, USA). In addition, Origin 2018 was used to fit the relationship between otolith length and total length.

## 3. Results

### 3.1. Ontogenetic Development

The collected eggs were mostly in the tailbud and caudal fin-appearing stage. After 5–8 h of incubation at 19–28 °C, the eggs began to develop otoliths. At this time, the auditory sac formed an oval shape and two otoliths (lapilli and sagittae) were visible inside. Following a 7–10 h incubation period, the eggs began to hatch. The Pre-FLS was from 0 to 3 dph. Nutrition was provided via yolk sac, and the notochord was not upturned. The FLS was from 3 to 6 dph. At this point, the notochord was upturned, and nutrition was provided via exogenous sources. Moreover, the caudal fin was almost completely formed, and the dorsal fin started to develop at 6 dph. The Post-FLS was from 6 to 13 dph, at which point, the pectoral, dorsal, and pelvic fins were visible. Lastly, the JS was from 14 to 25 dph, when the whole body became covered in scales. The transition age between the Pre-FLS and the FLS was at 4 dph, between the FLS and the Post-FLS at 6 dph, and between the Post-FLS and the JS at 13.5 dph. Otoliths of 2–3 groups were selected from the samples at each stage; the sample population, predicted otolith age (increment number), predicted daily developmental rate, and TLs were calculated. The data are presented in Table 1.

**Table 1.** Sampling details for each stage of larval and juvenile *Sinogastromyzon wui*, including the sample size (N), otolith age estimates (increment count), daily growth rate estimate (DGR), and TL.

| DPH | N | Increment Count (Mean ± SD, Days) | DGR (Mean ± SD, μm·day$^{-1}$) | TL (Mean ± SD, mm) |
|---|---|---|---|---|
| 2 | 10 | 1 ± 1 | 3.02 ± 0.26 | 6.02 ± 0.25 |
| 3 | 10 | 2 ± 1 | 2.64 ± 0.31 | 6.68 ± 0.26 |
| 4 | 10 | 3 ± 1 | 3.23 ± 0.49 | 6.75 ± 0.31 |
| 7 | 10 | 6 ± 2 | 4.49 ± 0.54 | 6.92 ± 0.40 |
| 13 | 10 | 12 ± 2 | 3.46 ± 0.58 | 8.46 ± 0.73 |
| 17 | 10 | 16 ± 2 | 4.99 ± 0.34 | 9.95 ± 0.67 |
| 23 | 10 | 22 ± 3 | 4.38 ± 0.43 | 11.31 ± 1.12 |

Only the lapilli and sagittae were visible after hatching, with the asterisci emerging at 4 dph. All otolith pairs underwent anatomical alterations during ontogenesis (Figure 2). Lapilli were round in shape at hatching (0 dph, TL = 6.05 ± 0.25 mm) and maintained their round shape from Post-FLS (13 dph, TL = 8.46 ± 0.73 mm) until the end of JS (25 dph, TL = 12.03 ± 0.92 mm), when the otolith developed into an ovoid.

The shape of the sagitta altered significantly during development. From the newly hatched larval stage to the Pre-FLS (0 dph, TL = 6.59 ± 0.28 mm), the shape of the sagitta changed from nearly round to oval. Subsequently, from the FLS (7 dph, TL = 6.92 ± 0.40 mm) until the Post-FLS (13 dph, TL = 8.46 ± 0.73 mm), it was shaped like a sickle. During the JS, the sagitta continued to develop but was more stable in terms of structure. The asteriscus was renal when it appeared (4 dph, TL = 6.68 ± 0.31 mm).

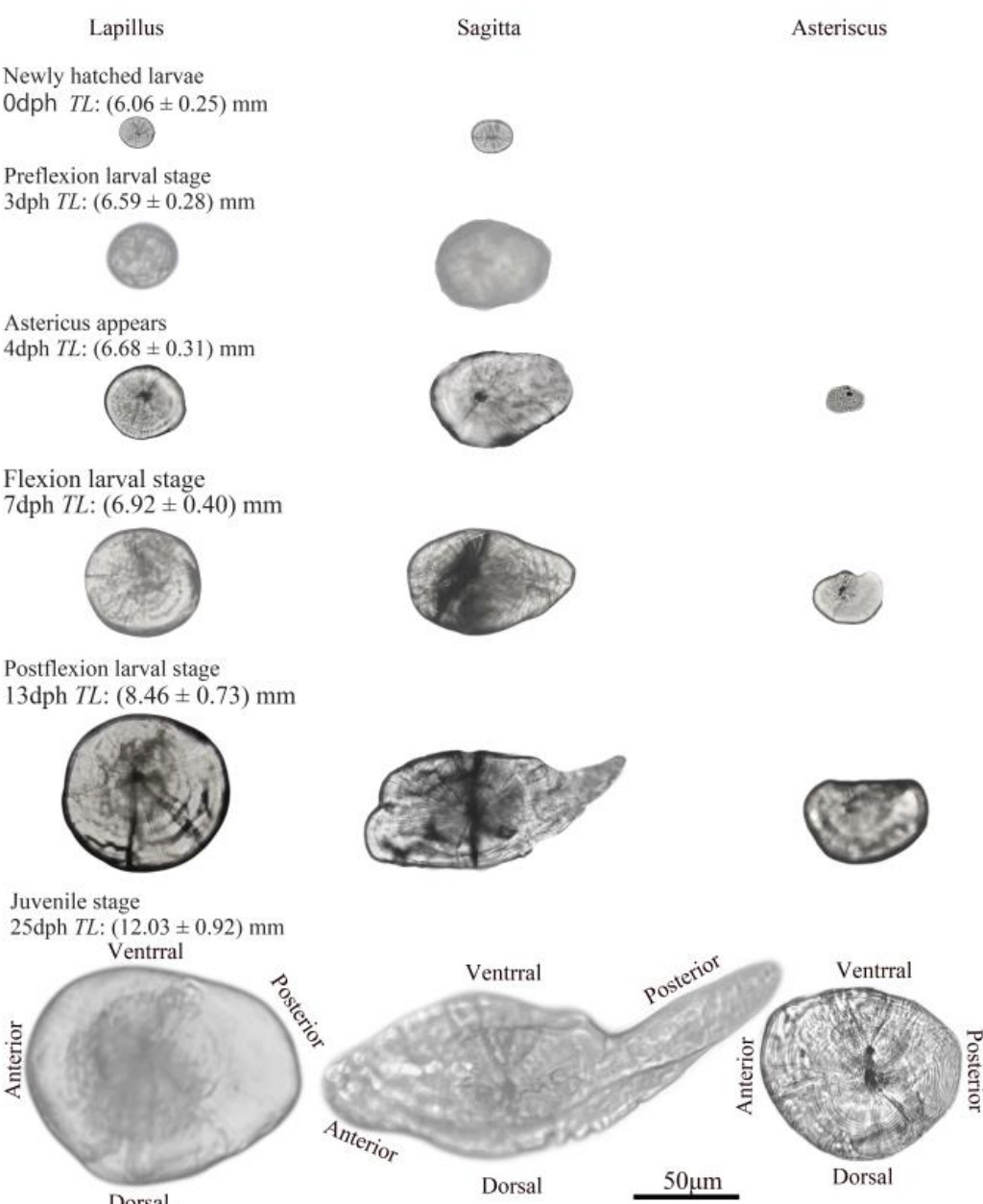

**Figure 2.** Otolith development in *Sinogastromyzon wui*.

### 3.2. Lapilli and Sagittae Shape Analysis

Our shape analyses demonstrated that both lapillus and sagitta structures underwent alterations with age. The age distinctions were more pronounced in sagittae than lapilli. Based on the lapilli CDA, the age-related changes were not distinguishable (Wilks' $\lambda > 0.05$; LD1 77.1%, LD2 13.5%) with the two discriminant functions of CDA (Figure 3a). In contrast, the sagittae CDA showed that the age distinctions were obvious (Wilks' $\lambda < 0.05$; LD1 98.3%) using the first discriminant function of CDA, which stratified ages into four classes (1 and 3 days; 5 days; 9, 13, and 17 days; 25 days) (Figure 3b).

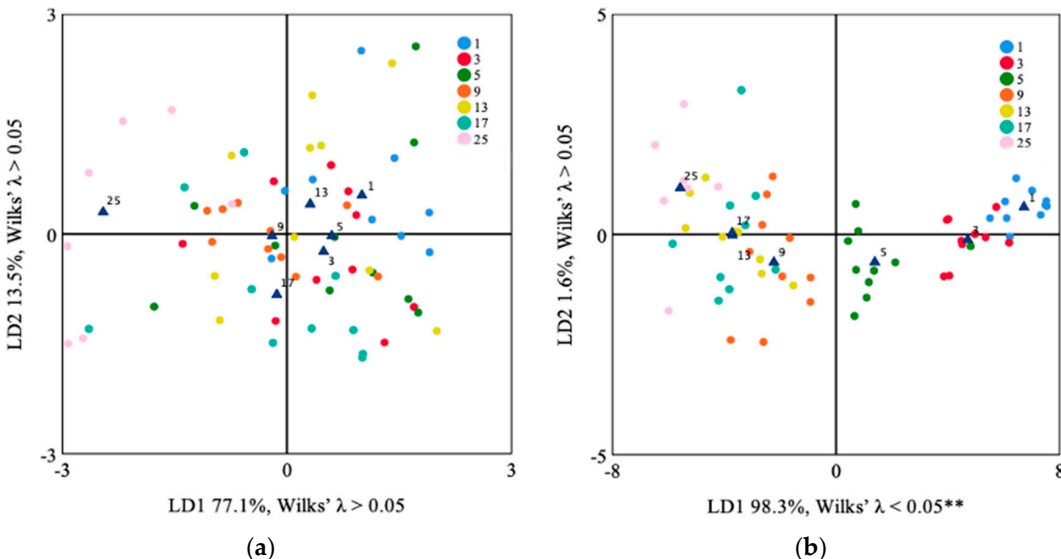

(a)  (b)

**Figure 3.** Canonical discriminant analyses performed on the Fourier coefficients of lapilli (**a**) and sagittae (**b**) from *Sinogastromyzon wui* of each age class. **, $p < 0.01$.

### 3.3. Otolith Growth

In this study, the logarithmic growth model displayed a very good fit for lapilli growth (Figure 4a), and the results showed no significant difference in the growth of each otolith area ($p > 0.05$). In contrast, the linear and gompertz growth model fit the sagitta growth remarkably (Figure 4b), and the growth in each area showed a significant difference ($p < 0.05$).

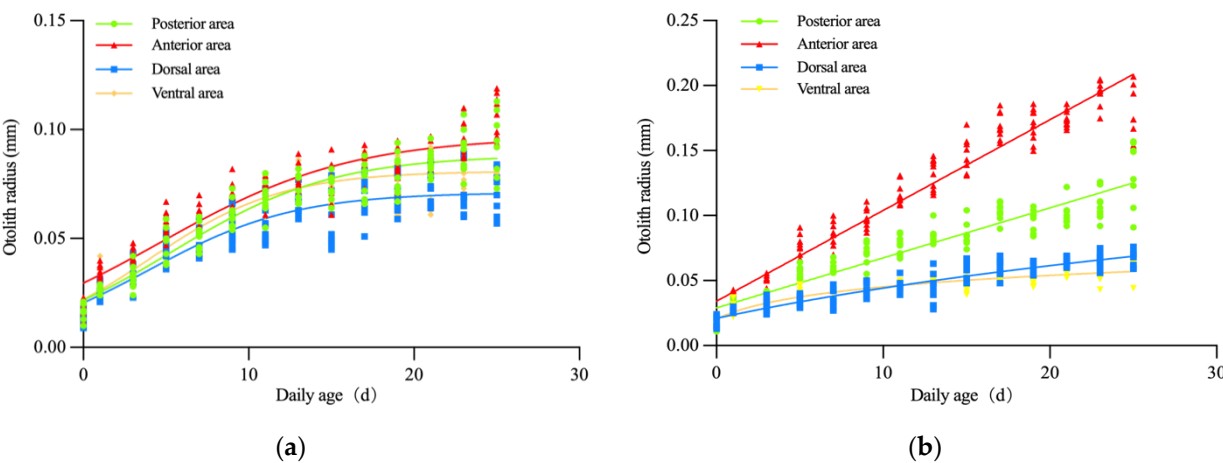

(a)  (b)

**Figure 4.** The relationship between daily age and otolith radius of lapilli (**a**) and sagittae (**b**) of *Sinogastromyzon wui*: (**a**) posterior area, $OR_t = -0.045 + 0.04 \times \ln(t + 4.563)\,(R^2 = 0.9086)$; anterior area, $OR_t = -0.013 + 0.033 \times \ln(t + 2.72)\,(R^2 = 0.9090)$; dorsal area, $OR_t = -0.001 + 0.022 \times \ln(t + 1.806)\,(R^2 = 0.8370)$; and ventral area, $OR_t = -0.0025 + 0.025 \times \ln(t + 1.631)\,(R^2 = 0.8299)$; (**b**) posterior area, $OR_t = 0.004t + 0.029\,(R^2 = 0.8838)$; anterior area, $OR_t = 0.0074t + 0.034\,(R^2 = 0.9449)$; dorsal area, $OR_t = 0.022e^{(-0.021x)}\,(R^2 = 0.8479)$; and ventral area $OR_t = 0.023e^{(-0.124x)}\,(R^2 = 0.8223)$.

The otolith length of lapilli ($OL_L$) was larger than the otolith length of sagittae ($OL_S$) at larval hatching, but $OL_S$ was larger than $OL_L$ when the body length of larvae reached $6.02 \pm 0.25$. Correlations between *TL* and $OL_L$ or $OL_S$ were fitted to the following exponential growth model: $OL_L = 4.494e^{4.694TL}$ ($R^2 = 0.786$, n = 150) (Figure 5) and $OL_S = 5.089e^{2.548TL}$ ($R^2 = 0.845$, n = 150) (Figure 5).

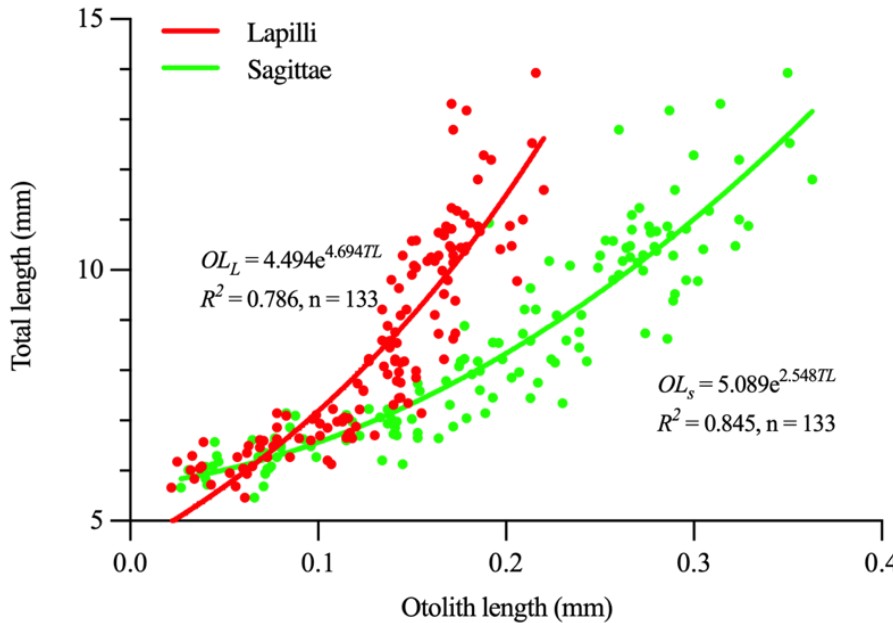

**Figure 5.** The relationship between otolith length of lapilli or sagittae and total length of *Sinogas-tromyzon wui*.

### 3.4. The Lapilli Otolith Microstructure

The otolith core area had one central primary primordia (Figure 6). Following incubation at 21–26 °C water temperature, the first daily increment of lapilli formed at 2 dph. The primordial core radius and the first developmental increment (F1) was between 11.0 and 14.4 μm in length, with an average of 13.5 ± 1.1μm (n = 10). Moreover, the width of the daily increment fluctuated. The daily increment width before Post-FLS (7–12 dph) widened (mean, <5.9 μm). However, past the breaking point, the increment width narrowed and then widened again as juveniles reached 20 dph, when it gradually reduced to the otolith edge (Figure 7).

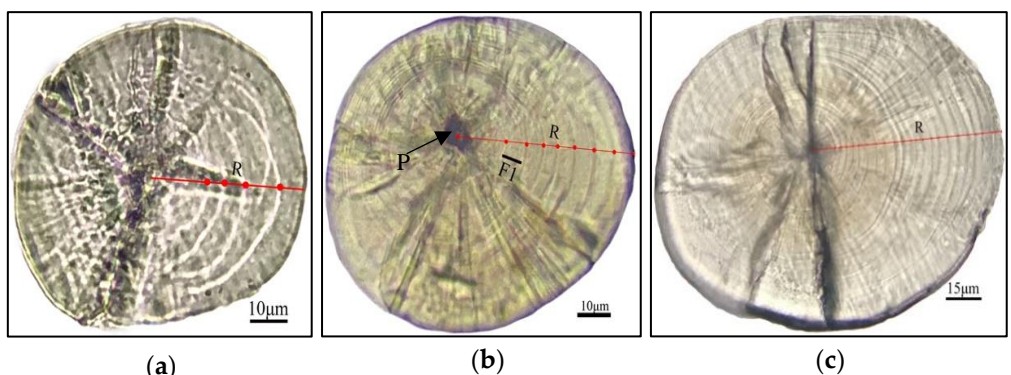

**Figure 6.** Lapilli otoliths of Sinogastromyzon wui. (**a**) The lapillus otolith of larva at 5 dph; showing the radius (R) along which increments were read and measured. (**b**) The lapillus otolith of larva at 9 dph, showing the primordium (P) and the first increment (F1). (**c**) The lapillus otolith of larva at 23 dph, showing the radius (R) along which increments were read.

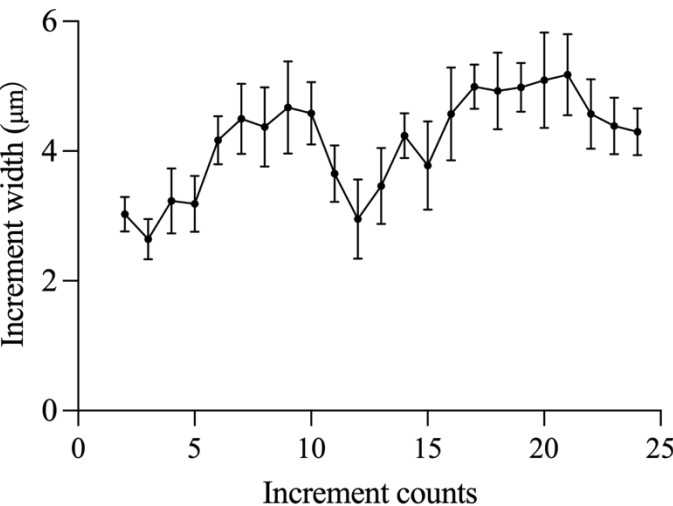

**Figure 7.** Daily increment widths (mean ± SD) of the lapilli.

*3.5. Verification of the Daily Lapilli Increment Formation*

Here, we verified the daily increment formation by testing 150 lapilli. To accomplish this, we fitted the correlation between growth increment numbers (N) and larval and juvenile ages (A) to the function as follows:

$$A = 0.97\,N - 1.08 \tag{1}$$

**4. Discussion**

This study revealed that the lapilli morphology did not significantly alter from the nearly round to ovoid shape, while the sagittae morphology underwent significant changes from nearly a round shape to a sickle. The otolith morphological development is similar to many reported Cypriniformes [26,29,30]. Moreover, otolith morphology is species-specific and is under the regulation of both genetic and environmental factors [31]. For instance, the lapilli of *Macropodus opercularis* develop from an oval shape in the larval stage to prismatic in the JS [32]. Moreover, the sagittae of *Liza haematocheila* develop from an oval shape in the larval stage to a leaf-like shape in the JS [33]. These data indicate that the otolith morphology in the JS exhibits a certain degree of species-specificity, which can be very helpful in species identification during the early stages of fish development [34].

The CDA results revealed no obvious correspondence between lapillus shape development and varying stages of larval and juvenile development. Lapilli were always nearly round during this period, which was consistent with the analysis results of the same growth rate in each lapillus region. However, in terms of the sagitta, there was a significant correlation between the shape development and varying stages of larvae and juveniles. During the Pre-FLS, the growth rate of each sagittal region was the same, and the shape remained round. During the FLS, the anterior and posterior regions began to grow faster than the ventral and dorsal regions; hence, the shape became an ellipse. During the Post-FLS, the anterior and posterior regions continued to grow rapidly, and the shape became a sickle. By the JS, the sagitta shape became quite stable. Therefore, the sagitta shape was more suitable than the lapillus shape for predicting the developmental stages of *S. wui* larvae and juveniles.

Herein, we made novel verifications of the otolith daily increment formation rate and initial increment development in *S. wui* larvae and juveniles. The first increment was obvious at 2 dph. This was consistent with the observation that *Myxocyprinus asiaticus* forms the first increment at 2 dph [29] but was different from other reports that stated as follows: *Oreochromis aureus* and *Cyprinus carpio* form the first increment before hatching [35,36]; *Acanthopagrus schlegeli* and *Theragra chalcogramma* form the first increment at hatching [37,38]; *Cirrhinus molitorella* forms the first increment at 3 dph [39]. The formation

of the first daily increment is species-specific, possibly due to the regulation of ontogenetic and environmental factors such as yolk absorption and first-feeding [39–42]. To avoid deviation in larval and juvenile age calculation, it is essential to confirm the initial daily increment deposition time separately [3]. In this study, we further validated the daily lapillus increment formation rates. We revealed that the larval and juvenile age can be calculated by D + 1, which is consistent with reports of most fish [39,43]. The conclusions from this study can be used for the reproductive duration and early growth study of *S. wui*.

The microstructures and growth features of otolith should be extensively studied to better elucidate the critical events and stages of larvae and juveniles [23]. Among the previously examined factors are hatch check [44], first-feeding [35], and development of numerous accessory growth centers [45]. In this study, we only detected one growth center. There was no clear hatching check or sign of first-feeding. We observed fluctuating alterations in daily increment width. For example, the width gradually increased during the FLS and the early JS, which corresponded to the exogenous and active ossification transitions, respectively. Based on other studies, in terms of growth, the association between body length (BL) and otolith radius (OR) can be either linear [45,46] or nonlinear [26,35]. In this study, an exponential growth curvilinear was employed to fit the BL–OL correlation. We observed a shift in the relationship at the end of the larval stage (OL, 0.20–0.25 mm). The same result was also observed in *Cirrhinus molitorella* [35], *Siniperca chuatsi* [45], *Strangomera bentincki,* and *Engraulis ringens* [47], indicating the likelihood that this shift is brought on by endogenous factors such as genetics, onto-genetics, and physiology.

## 5. Conclusions

These results provide previously unavailable ontogeny and otolith growth information for *S. wui* that can increase an understanding of its biology. In the Xijiang River, the results of this study can be used to assess the dynamics of *S. wui* recruitment and the relationship between their ontogeny and key environmental factors such as river flow, temperature, and feed, which may provide insights into the relationship between changes in river environment and overall population dynamics. Moreover, as *S. wui* is the dominant species in this river, the migration of its spawning grounds may better reveal the impact of dam closure on fish spawning in the upstream section of the river.

**Author Contributions:** Conceptualization, methodology, software and writing—original draft preparation, M.G.; formal analysis and validation X.T.; investigation, H.H.; resources and data curation, M.L.; writing—review and editing, supervision, Z.W.; project administration and funding acquisition, L.H. All authors have read and agreed to the published version of the manuscript.

**Funding:** This research was funded by the National Natural Science Foundation of China, grant number "32060830" and "U20A2087".

**Institutional Review Board Statement:** The study was conducted according to the guidelines of the Declaration of Helsinki and approved by the Institutional Review Board of Animal Experimental Ethics committee of Guangxi University (GXU-2022-1 and 10 April 2021).

**Data Availability Statement:** Data from this study are available from the corresponding author upon request (Z.W.: wuzhiqiang@glut.edu.cn).

**Acknowledgments:** We are grateful to the "Chunhui Planning" Project from the Ministry of Education, China. We also thank all the editors and reviewers for providing constructive comments on the present work.

**Conflicts of Interest:** The authors declare no conflict of interest.

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
