# Peer review of "Growth and Microstructural Features in Otoliths of Larval and Juvenile Sinogastromyzon wui (F. Balitoridae, River Loaches) of the Upper Pearl River, China"

_fishes, doi:10.3390/fishes7020057_

Round 1

Reviewer 1 Report

Dear Authors,

I found your manuscript interesting, quite innovative, and well focused on the research question. The otoliths study represent an innovative way to investigate for a better comprehension of some issue related to teleost ant their biology and ecology. You focused on a little studied species and on some partially unknown early life stages ontogeny, starting from the eggs, good choice.

Despite this manuscript captured my interest, I have an overall recommendation to enhance it, English language needs a revision by an expert to give more fluency and improve the scientific soundness of this document. Of course, I suggest you to submit the document to the language editing after the corrections and the partial rewriting of the manuscript in relation to the others following requests.

Moreover, I have some other minor revision to address before reconsider this manuscript for publication in Fishes Journal, that I briefly summarize as follow.

Title

Is better to give the name of the discoverer of the species, in this case (Fang, 1930).

Keywords

To give more resonance during the web searches of other researchers, is better to avoid the repetition of words already reported in Title, as in your case: 

Larval and juvenile; Otolith; Microstructural features;

Please try to substitute it with others related, like for example: ontogeny; teleost anatomy; fish morphology;

Introduction

Introduction section appears poor in the present form, strictly related only to the essential parts of this big topic, selective treated by the Authors during the study. This is of course essential, to develop the main features treated in you study, but this topic needs in my opinion a more wide introduction on the importance of otoliths nowadays and the various applications linked to their study. Please try to enriching this section, particularly in its first general part, using some reference like the following (all useful for the Discussion too):

D’Iglio, C.; Albano, M.; Famulari, S.; Savoca, S.; Panarello, G.; Di Paola, D.; Perdichizzi, A.; Rinelli, P.; Lanteri, G.; SpanoÌ€, N.; et al. Intra- and interspecific variability among congeneric Pagellus otoliths. Sci. Rep. 2021, 11, 16315.

Popper, A.N.; Ramcharitar, J.; Campana, S.E. Why otoliths? Insights from inner ear physiology and fisheries biology. Mar. Freshw. Res. 2005, 56, 497–504.

Zhuang, L.; Ye, Z.; Zhang, C.; Ye, Z.; Li, Z.; Wan, R.; Ren, Y.; Dou, S.; Wheeler, A.; Whitehead, P.J.P.; et al. Stock discrimination of two insular populations of diplodus annularis (Actinopterygii: Perciformes: Sparidae) along the coast of tunisia by analysis of otolith shape. J. Fish. Biol. 2015, 46, 1–14. 

D’Iglio, C.; Natale, S.; Albano, M.; Savoca, S.; Famulari, S.; Gervasi, C.; Lanteri, G.; Panarello, G.; SpanoÌ€, N.; Capillo, G. Otolith Analyses Highlight Morpho-Functional Differences of Three Species of Mullet (Mugilidae) from Transitional Water. Sustainability 2022, 14, 398.

Bose, A.P.H.; Zimmermann, H.; Winkler, G.; Kaufmann, A.; Strohmeier, T.; Koblmüller, S.; Sefc, K.M. Congruent geographic variation in saccular otolith shape across multiple species of African cichlids. Sci. Rep. 2020, 10, 1–14. 

Material and Methods

Line 98: please add the name of the utilized software or image analysis system.

Results

In 2.2 line 85 the Authors report the use of 320 samples in their study, but in Table 1 were reported only 70 samples in total, can the Authors better expose the experimental and sampling design?

Discussion

Line 221: please support this sentence with some references.

Line 226: please support this sentence with some references.

Line 238: please better argue this topic with similar studies and support this sentence with some references.

Line 249: please support this sentence with some references.

Line 253: such as for line 238, this topic needs to be deeply treated and compared with other similar studies.

Conclusion 

This section was avoided, I think by choice, by the Authors, but the key results of this good study were dispersed or missed within Discussion section. Please add this separate section or rewrite the final part of Discussion to summarize the obtained results, to give more resonance and scientific soundness to your good work.

Have a nice work.

The Reviewer

Author Response

We would like to express our sincere thanks to the reviewers for the constructive and positive comments.

Replies to Reviewer 1

Title

Is better to give the name of the discoverer of the species, in this case (Fang, 1930).

Answer: Correction has been made in the revised version.

Keywords

To give more resonance during the web searches of other researchers, is better to avoid the repetition of words already reported in Title, as in your case:

Larval and juvenile; Otolith; Microstructural features;

Please try to substitute it with others related, like for example: ontogeny; teleost anatomy; fish morphology;

Answer: Correction has been made in the revised version.

Introduction

Introduction section appears poor in the present form, strictly related only to the essential parts of this big topic, selective treated by the Authors during the study. This is of course essential, to develop the main features treated in you study, but this topic needs in my opinion a more wide introduction on the importance of otoliths nowadays and the various applications linked to their study. Please try to enriching this section, particularly in its first general part, using some reference like the following (all useful for the Discussion too):

D’Iglio, C.; Albano, M.; Famulari, S.; Savoca, S.; Panarello, G.; Di Paola, D.; Perdichizzi, A.; Rinelli, P.; Lanteri, G.; Spanò, N.; et al. Intra- and interspecific variability among congeneric Pagellus otoliths. Sci. Rep. 2021, 11, 16315.

Popper, A.N.; Ramcharitar, J.; Campana, S.E. Why otoliths? Insights from inner ear physiology and fisheries biology. Mar. Freshw. Res. 2005, 56, 497–504.

Zhuang, L.; Ye, Z.; Zhang, C.; Ye, Z.; Li, Z.; Wan, R.; Ren, Y.; Dou, S.; Wheeler, A.; Whitehead, P.J.P.; et al. Stock discrimination of two insular populations of diplodus annularis (Actinopterygii: Perciformes: Sparidae) along the coast of tunisia by analysis of otolith shape. J. Fish. Biol. 2015, 46, 1–14.

D’Iglio, C.; Natale, S.; Albano, M.; Savoca, S.; Famulari, S.; Gervasi, C.; Lanteri, G.; Panarello, G.; Spanò, N.; Capillo, G. Otolith Analyses Highlight Morpho-Functional Differences of Three Species of Mullet (Mugilidae) from Transitional Water. Sustainability 2022, 14, 398.

Bose, A.P.H.; Zimmermann, H.; Winkler, G.; Kaufmann, A.; Strohmeier, T.; Koblmüller, S.; Sefc, K.M. Congruent geographic variation in saccular otolith shape across multiple species of African cichlids. Sci. Rep. 2020, 10, 1–14.

Answer: The references has been added in the revised version.

Material and Methods

Line 98: please add the name of the utilized software or image analysis system.

Answer: The name of the utilized software or image analysis system (computer imaging identification system for the analysis of otolith microstructure of fish) has been added in the revised version (Line 114).

Results

In 2.2 line 85 the Authors report the use of 320 samples in their study, but in Table 1 were reported only 70 samples in total, can the Authors better expose the experimental and sampling design?

Answer: Otolith samples were selected daily during Pre-FLS and FLS, following that otolith samples were selected once every 2 days during Post-FLS and JS, total 320 individuals (10 in every age group used for shape research, and 10 for validating increment deposition frequency). 2-3 groups were selected from samples at each stage sample population, predicted otolith age (increment number), predicted daily developmental rate, and otolith TLs were calculated. The data are presented in Table 1.

Discussion

Line 221: please support this sentence with some references.

Line 226: please support this sentence with some references.

Line 238: please better argue this topic with similar studies and support this sentence with some references.

Line 249: please support this sentence with some references.

Line 253: such as for line 238, this topic needs to be deeply treated and compared with other similar studies.

Answer: The references have been added in the Discussion. And the topics have been deeply treated.

Conclusion 

This section was avoided, I think by choice, by the Authors, but the key results of this good study were dispersed or missed within Discussion section. Please add this separate section or rewrite the final part of Discussion to summarize the obtained results, to give more resonance and scientific soundness to your good work.

Answer: The conclusion has been added in the revised version.

Reviewer 2 Report

This paper present one study on the growth and microstructural features of otoliths in larval and juvenile Sinogastromyzon wui. This is a very good scientific idea around the 3 differents otoliths (lapillus, sagitta and astericus). Consequently, this paper is a very good contribution from scientific research. The growth and the shape of otoliths are very important in fisheries science.

However, the method presented in this paper must be improved to answer to different questions :

1) how do the shape of the 3 otoliths evolve over time and also between them?   

to answer this question, it is necessary to use the fourier descriptors to compare the differences (% overlap) over time and between the 3 otoliths on each sampling day.   

To visualise shape differences between 2 groups of otoliths (example sagitta at 3 dph and sagitta at 4 dph, sagitta at 3 dph vs astericus at 3 dph), average shapes were rebuilt based on EFDs averaged for each group and the perecentage of overlapp could be estimated.

one example of paper with this information : https://doi.org/10.1016/j.jembe.2019.151239

2) to compare the growth from each of the 3 otoliths and have an analysis of the relationship between the different otoliths 

This paper showed that the relationship of each otolith with the total lenght. it will be important to show the relationship between different otoliths

3) within each pair of otoliths, it is possible to test the difference between left and right otoliths : to know if it possible to use the left or right otolith without bias   

By integrating these 3 questions with your data requiring additional analysis, your paper will be much more interesting scientifically. 

Another small points :

  • Figure 1 : pleasae identify the color segment as Figure 3 to better understand
  • paragraph 3.2 : this analysis showed the diffrents around the time but what are the difference ?  You could add the shape indices (rectangularity....) to identify the difference
  • Figure 4 : only 1 figure will be better to compare the difference between sagittae and lapillii
  • for larvae otoliths, the polishing technic is often used to better see the increment because the increments in your figure 5a are "bizzard"

Author Response

We would like to express our sincere thanks to the reviewers for the constructive and positive comments.

Replies to Reviewer 2

1) how do the shape of the 3 otoliths evolve over time and also between them?   

to answer this question, it is necessary to use the fourier descriptors to compare the differences (% overlap) over time and between the 3 otoliths on each sampling day. 

Answer: This is a very good question, it helps us clear daily otolith shape changes,  belongs to the analysis of otolith growth. In this study, the corresponding relationship between the shape change of otoliths and the development stages of larvae and juveniles was first studied, the growth of the otolith preceded by analyzing the relationship between each area of otolith radius and dph.

2) to compare the growth from each of the 3 otoliths and have an analysis of the relationship between the different otoliths 

Answer: The relationship between the lapillus and sagitta has been analyzed

in the revised version. We excluded asteriscus from the analysis due to its note present at hatching.

3) within each pair of otoliths, it is possible to test the difference between left and right otoliths: to know if it possible to use the left or right otolith without bias   

Answer: Due to the highly fragile nature of agitate,  in this study,all intact otoliths were used in the analysis.

Another small points :

Figure 1: please identify the color segment as Figure 3 to better understand

paragraph 3.2 : this analysis showed the diffrents around the time but what are the difference ?  You could add the shape indices (rectangularity....) to identify the difference

Figure 4 : only 1 figure will be better to compare the difference between sagittae and lapillii

for larvae otoliths, the polishing technic is often used to better see the increment because the increments in your figure 5a are "bizzard"

Answer: Correction has been made in the revised version.

Reviewer 3 Report

The study depicted in the paper is sound and interesting to those who are involved in age and growth of fish and inner ear morphology of fishes. There are a few grammatical issues that should be addressed (see attached pdf). The most significant change I think is needed is a some introductory material on the morphology of otoliths, particularly the sagittae, in the Ostariophysi subocohort of fishes to which this species belongs. Bringing this information up in the introduction will better set the stage for the study that was conducted and the conclusions that were reached.

Author Response

We would like to express our sincere thanks to the reviewers for the constructive and positive comments.

Replies to Reviewer 3

Correction has been made in the revised version.

Round 2

Reviewer 1 Report

Dear Authors, 

I'm glad to see that all my previous comments and suggestion on your manuscript have been useful to improve your manuscript. Thank you for the clarifications about experimental design. The manuscript is now enhanced and ready for publication in the present form, is more well organized with some important topic more in-depth treated in Introduction and Discussion sections, more detailed starting from Title and Keywords, and your key results has found the right soundness in the added Conclusion section.

Well done!

Best regards

The Reviewer

Author Response

We would like to express our sincere thanks for your constructive and positive comments.

Reviewer 2 Report

Dear authors,

before to accept this paper two small modifications are important :

1) in the title, you must identified the geographical area because the growth is depend on the geographical area  

2) the figure 1 showed the measures on the otoliths but in all papers on the otolith measures, the width segment is always perpendical to the otolith length to limit the bias of user. therefore, it would be better to show the perpendicular segments if it is a true

best regards

Author Response

We would like to express our sincere thanks for your constructive and positive comments.

Replies to Reviewer 2

1) in the title, you must identified the geographical area because the growth is depend on the geographical area 

Answer: Correction has been made in the revised version.

2) the figure 1 showed the measures on the otoliths but in all papers on the otolith measures, the width segment is always perpendical to the otolith length to limit the bias of user. therefore, it would be better to show the perpendicular segments if it is a true

Answer: Correction has been made in the revised version.
